# Association between Ambulatory Status and Functional Disability in Elderly People with Dementia

**DOI:** 10.3390/ijerph16122168

**Published:** 2019-06-19

**Authors:** Hsun-Hua Lee, Chien-Tai Hong, Dean Wu, Wen-Chou Chi, Chia-Feng Yen, Hua-Fang Liao, Lung Chan, Tsan-Hon Liou

**Affiliations:** 1Department of Neurology, Shuang Ho Hospital, Taipei Medical University, New Taipei City 23561, Taiwan; kaorulei@yahoo.com.tw (H.-H.L.); chientaihong@gmail.com (C.-T.H.); tingyu02139@gmail.com (D.W.); 2Dizziness and Balance Disorder Center, Shuang Ho Hospital, Taipei Medical University, New Taipei City 23561, Taiwan; 3Department of Neurology, School of Medicine, College of Medicine, Taipei Medical University, Taipei 11031, Taiwan; 4Taiwan Society of International Classification of Functioning, Disability and Health, TSICF, New Taipei City 23561, Taiwan; y6312002@gmail.com (W.-C.C.); mapleyeng@mail.tcu.edu.tw (C.-F.Y.); hfliao@ntu.edu.tw (H.-F.L.); 5Department of Occupational Therapy, Chung Shan Medical University, Taichung 40201, Taiwan; 6Department of Public Health, Tzu Chi University, Hualien City 97004, Taiwan; 7School and Graduate Institute of Physical Therapy, College of Medicine, National Taiwan University, Taipei 10617, Taiwan; 8Department of Physical Medicine and Rehabilitation, Shuang Ho Hospital, Taipei Medical University, New Taipei 23561, Taiwan; 9Department of Physical Medicine and Rehabilitation, School of Medicine, College of Medicine, Taipei Medical University, Taipei 11031, Taiwan; 10Graduate Institute of Injury Prevention and Control, College of Public Health, Taipei Medical University, Taipei 11031, Taiwan

**Keywords:** dementia, gait, World Health Organization Disability Assessment Schedule 2.0 (WHODAS 2.0), International Classification of Functioning, Disability and Health (ICF)

## Abstract

Dementia is highly comorbid with gait disturbance, and both conditions negatively impact the ability of elderly people to conduct daily living activities. The ambulatory status of older adults with dementia may cause variable functional disability, which is crucial for the progression of dementia. The present study investigated the association between ambulatory status with functional disability in elderly people and dementia by using the World Health Organization Disability Assessment Schedule 2.0 (WHODAS 2.0). In total, 34,040 older adults with mild-to-advanced dementia were analyzed and categorized according to their ambulatory status into three groups: Nonambulatory, assisted ambulatory, and ambulatory. In general, poor ambulatory status was associated with both greater severity of dementia and functional disability. The study participants were further segregated according to their ages and dementia severity levels, which demonstrated that the WHODAS 2.0 domains of functioning for getting along, life activities, and participation (domains 4, 5–1, and 6) were all associated with ambulatory status. In addition, nonambulatory status was significantly associated with institution residency among older adults with dementia. In conclusion, the present study clearly demonstrated the role of ambulatory status in functional disability in older adults with dementia, and the association persisted among older adults of different ages and severities of dementia. This finding indicates the importance of maintaining walking ability in the management of dementia in older adults.

## 1. Introduction

Dementia is a progressive degenerative disorder characterized by a decline in cognition. It is one of the most critical health concerns worldwide. Approximately 46 million people worldwide have dementia [1]. Dementia contributes to 11.2% of years lived with disability in people aged 60 years and older [2]. The major symptoms of dementia are multiple cognitive impairments, including memory, executive function, learning, language, attention, perceptual-motor, and social cognition impairments [3]. In addition to these well-known problems, people with dementia also experience other comorbidities, such as gait disturbance or slowing, depression, and abnormal eye movements, all of which can also affect quality of life [4,5].

Gait is a motor behavior that requires coordination between multiple nervous systems, including the sensorimotor, extrapyramidal, and cerebrocerebellar systems [6]. Aging is associated with several gait abnormalities, such as slower gait speed, increased stride width, shorter stride length, and prolonged double-limb support time [7]. Among older adults, gait impairment is more common in those with dementia, which thereby increases the risk of falls, fractures, and even mortality [8,9]. Gait abnormalities can be observed in any stage of dementia, even early stages [10]. Several studies have found that the slowing of gait and gait dysrhythmia may precede cognitive impairment and dementia [11,12]. Impaired executive function also leads to decreased walking speed, increased variability of stride time, increased incidence of falls, and decreased performance on complex motor tasks [13]. Gait disturbance further worsens disability in those with both dementia and advanced age [14], which may further promote the progression of dementia.

The World Health Organization Disability Assessment Schedule 2.0 (WHODAS 2.0) is a detailed tool that is based on any medical disease, psychiatric illness, or comorbid condition for measuring functioning and disability [15]. The WHODAS 2.0 comprises six major functional domains of life: Cognition, getting around, self-care, getting along, life activities, and participation [16]. No large-scale study has to date used the WHODAS 2.0 to evaluate disability in older adults with dementia. The present study therefore investigated the functional disability of older adults with dementia in terms of their ambulatory status to identify the effect of ambulatory status on functional disability in older adults with dementia.

## 2. Materials and Methods

### 2.1. Participants

The present study collected data from the Taiwan Databank of Persons with Disabilities for the period between July 2012 and October 2018. This study was approved by the Joint Institutional Review Board of Taipei Medical University (N201805048).

Participant selection is presented in Figure 1. In brief, patients with dementia were identified who had completed assessment, including the Clinical Dementia Rating (CDR) scale, WHODAS 2.0, and ambulatory assessment. Dementia was diagnosed according to the International Classification of Diseases, Ninth Revision, Clinical Modification (ICD-9-CM) and ICD-10-CM, by using the following disease codes: ICD-9-CM codes: 290.0, 290.1, 290.2, 290.3, 290.4, 294.1, 331.0, 331.1, 331.2, 331.7, 331.8, and 331.9; ICD-10-CM codes: F01.50, F01.51, F02.80, F02.81, F03.90, G13.8, G30.0, G30.1, G30.8, G30.9, G31.01, G31.09, G31.1, G31.2, G31.83, G31.85, G31.89, and G31.9. Individuals who failed to report details regarding education level, who did not complete all questions in the WHODAS 2.0, or who were younger than 65 years were excluded. Ultimately, a total of 34,040 participants with dementia were included in the analysis.

### 2.2. Research Tools

#### 2.2.1. Clinical Dementia Rating

The CDR, a global rating scale, was developed for a prospective study on mild dementia of the Alzheimer type. Its reliability, validity, and correlational data have been examined previously. Six cognitive categories are involved in this international standard for the staging of dementia, namely memory, orientation, judgment and problem solving, community affairs, home and hobbies, and personal care. The CDR was found to unambiguously distinguish between severely impaired and healthy older adults [17].

#### 2.2.2. WHODAS 2.0

The 36-item Chinese version of WHODAS 2.0 was utilized [18]. The questionnaire comprises six domains. Domain 1 assesses cognition, which involves communication and thinking activities, including concentration, memory, problem solving, learning, and communicating. Domain 2 assesses mobility. Activities, such as standing, moving around inside the home, leaving the home, and walking a long distance, are included in this domain. Domain 3 assesses the ability of self-care, such as hygiene, dressing, eating, and staying alone. Domain 4 assesses the ability to get along, including difficulties in interacting with others due to a health condition. Domain 5 assesses difficulty in performing day-to-day activities, including domestic responsibilities and activities during leisure and at work or school. Domain 6 evaluates social participation, such as community activities, barriers and hindrances to participating, and other issues, such as maintaining personal dignity.

#### 2.2.3. Ambulatory Status

Ambulatory status was evaluated by asking the individuals to walk 3 meters and then return, which was included in the Taiwan Databank of Persons with Disabilities. A score of 0 represented the need for ambulatory care, 1 to 3 represented the need for assisted ambulatory care, and 4 represented no need for ambulatory care.

### 2.3. Statistical Analysis

Statistical analysis was performed using SAS Statistical Analysis System (version 9.2; SAS Institute Inc., Cary, NC, USA). The chi-squared test was used to analyze demographic variables, including sex, age, education, residence, urbanization level, and CDR. One-way analysis of variance with post hoc analysis was used to compare the differences in the WHODAS 2.0 scores. Logistic regression models were used to calculate odds ratios (ORs) and 95% confidence intervals (CIs) for walking status and each component factor according to the CDR. A *p* value of <0.05 was considered statistically significant.

## 3. Results

The demographic characteristics of the study participants are presented in Table 1. Overall, 34,040 people with mild-to-advanced dementia completed the ambulatory assessment. According to their walk status, they were divided into three groups: Ambulatory, assisted ambulatory, and nonambulatory. Women were predominant in all three groups. Regarding the age distribution, people aged 75 to 84 years were the majority in all three groups. A significant difference was noted in the education level, urbanization level, and severity of dementia. The institution residency rate was low (4.8%) in the ambulatory group and 38.8% in the nonambulatory group. A similar pattern was identified in all six domains of the WHODAS 2.0.

Considering the interaction between age, cognition, and ambulatory status and its effects on life activity, the present study subgrouped the study participants based on age, severity of dementia, and ambulatory status to compare the functional disability assessed using WHODAS 2.0. Instead of analyzing all six domains, we focused on the getting along (domain 4), life activities (domain 5), and participation (domain 6) domains because these functions are complex and integrated, whereas the cognition (domain 1), mobility (domain 2), and self-care (domain 3) domains are highly associated with one or two factors. As shown in Figure 2 (and the full descriptive statistics in Appendix A
Table A1), among the different age groups (65–74, 75–84, and ≥85 years), poorer ambulatory status was significantly associated with greater functional disability in those with mild, moderate, and severe dementia.

We further analyzed the association between ambulatory status and these three domains by using a logistic regression model (Table 2). Among people with dementia, older age and higher severity of dementia were significantly associated with nonambulatory status. Notably, nonambulatory status was significantly associated with institution residency (OR = 3.809, 95% CI, 3.576–4.058, *p* < 0.001) after adjusting for age, sex, and severity of dementia.

## 4. Discussion

This was the first study to investigate the association between ambulatory status and dementia severity among older adults based on the WHOAS 2.0. We found that ambulatory status was associated with dementia severity. Among people of different ages with different severities of dementia, ambulatory status further affected functional disability. For older adults with dementia, nonambulatory status was significantly associated with the risk of institution residency.

Gait disturbance is frequently seen in elderly people and patients with neurological disease [19]. Patients with dementia exhibit not only cognitive function impairment but also gait abnormality, especially those with executive function impairment [20]. Moreover, poor navigation, visuospatial perception, or attention can cause walking difficulties, immobility, and disability [21]. Several studies have investigated the correlation between gait and dementia by using different tools; for example, dual-task analysis [22]. WHODAS 2.0 is widely used for assessing function and activity with favorable reliability and validity [23]. In patients with dementia, a study showed that WHODAS 2.0 helps determine the risk of institutionalization [24]. The present study was the first to use WHODAS2.0 for assessing walking function and disability among patients with dementia.

Our study found that the severity of dementia was a valuable predicting marker associated with walking status. The appearance of abnormalities of gait increase the risk of dementia. Both of them were common with increasing age, making these two problems a major issue of public health. Patients with dementia with a higher CDR were associated with nonambulatory status in the getting along, life activities, and participation domains. Thus, prevention of dementia progression may help maintain walking status in patients with dementia. Furthermore, maintaining walking status and preventing nonambulatory status in patients with dementia can effectively improve cognitive function [25,26]. How to maintain the function of gait and decrease the severity of dementia is an urgent problem and requires more research.

Some limitations of our study must be discussed. First, this was a cross-sectional study, and hence, following up on the ability of patients to maintain walking status with good function was difficult. Second, the study analyzed disability in Taiwanese people with dementia, and thus, we did not analyze patients with mild cognitive impairment. Patients with mild cognitive impairment rarely experience any restriction in performing daily activities. Third, the low ethnic diversity of the Taiwanese population and differences in Taiwan’s health care system may have led to differences in the evaluated walking status among the studied population. Taiwan’s health insurance, medical, and social welfare systems are relatively distinctive. In addition, elderly people in Taiwan mostly live in the family home; therefore, the results of this study may not be applicable to patients in other countries.

## 5. Conclusions

Our research provides clinical features for elderly people with gait disorders and dementia. The quality of the gait can be sufficiently reliable and responsive to measuring the severity of dementia. Poor gait ability was associated with a higher CDR and WHODAS 2.0 score in individuals with dementia. Nonambulatory status was associated with both greater severity of dementia and functional disability. Further research involving longitudinal studies is certainly warranted to investigate the causes in detail.

## Figures and Tables

**Figure 1 ijerph-16-02168-f001:**
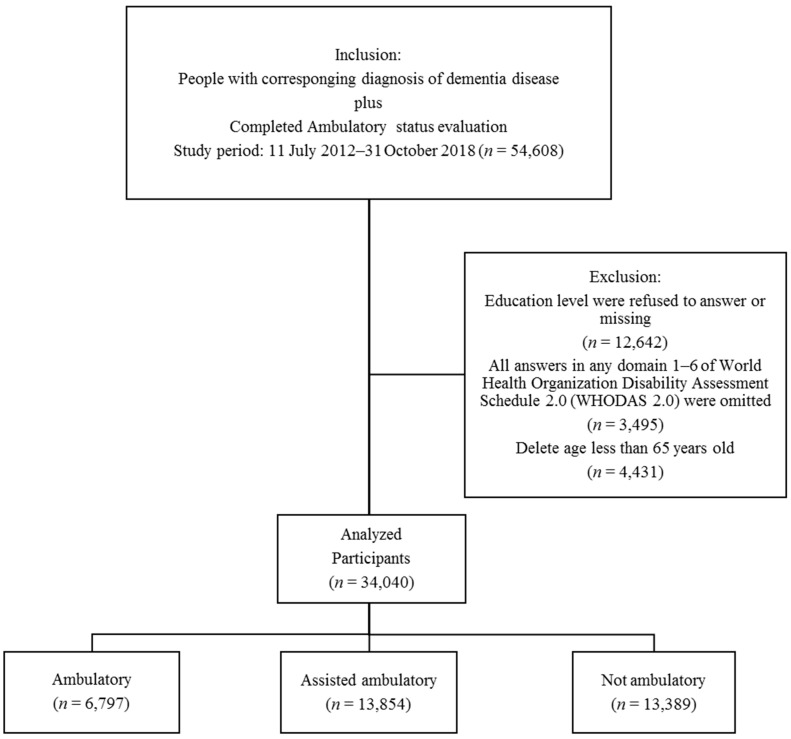
Flow chart of participant selection. In total, 34,040 demented participants with different ambulatory status were included in the analysis.

**Figure 2 ijerph-16-02168-f002:**
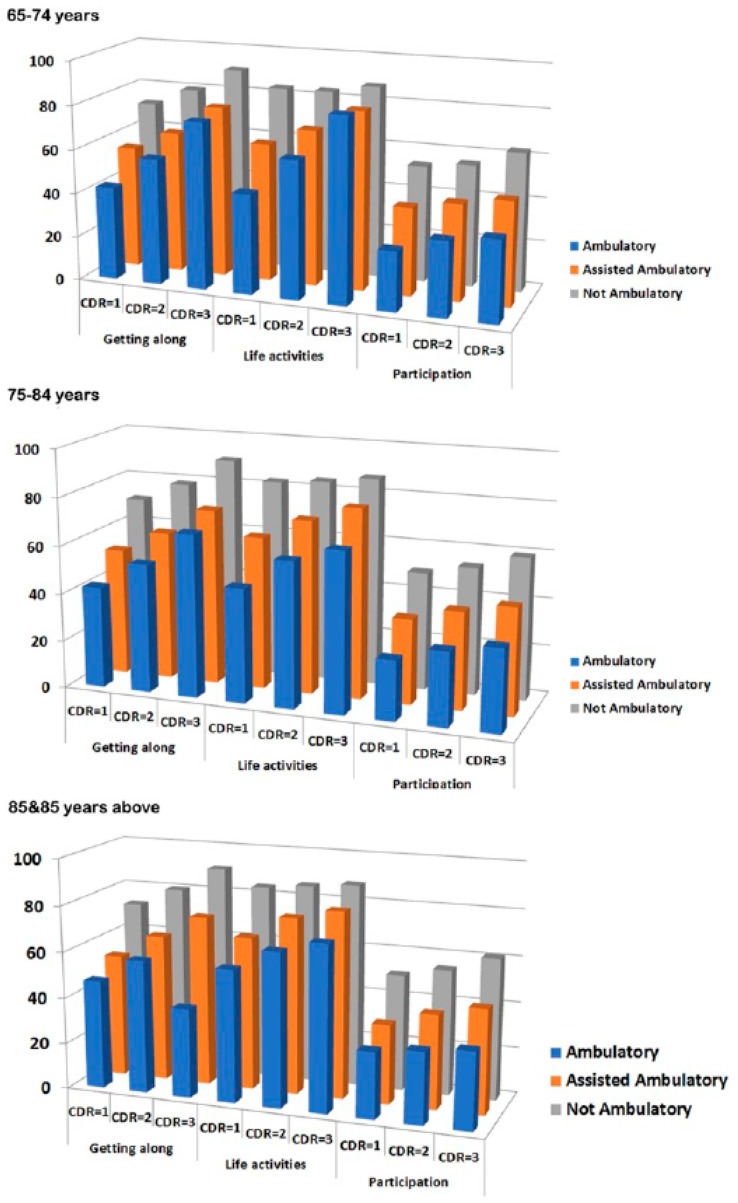
Different domains of World Health Organization Disability Assessment Schedule 2.0 (getting along, life activities, and participation) with ambulatory status and Clinical Dementia Rating in different age group (65–74, 75–84, ≥85 years).

**Table 1 ijerph-16-02168-t001:** Demographic data of all study participants.

	Ambulatory (*n* = 6797)	Assisted Ambulatory (*n* = 13,854)	Nonambulatory (*n* = 13,389)	*p*-Value
Sex (*n*, %)				0.5100
Male	2592, 38.1%	5398, 39.0%	5192, 38.8%	
Female	4205, 61.9%	8456, 61.0%	8197, 61.2%	
Age (years) (*n*, %)				<0.0001
65–74	2413, 35.5%	3348, 24.2%	2545, 19.0%	
75–84	3969, 58.4%	8618, 62.2%	7778, 59.1%	
≥85	415, 6.1%	1888, 13.6%	3066, 22.9%	
Total (mean ± SD)	76.8 ± 5.7	79.0 ± 6.1	80.6 ± 6.6	<0.0001
Education level (*n*, %)				<0.0001
Above college	54, 0.8%	132, 1.0%	133, 1.0%	
Senior high	315, 4.6%	663, 4.8%	606, 4.5%	
Junior high	264, 3.9%	578, 4.2%	617, 4.6%	
Primary	5217, 76.8%	9712, 70.1%	8092, 60.4%	
Illiterate	947, 13.9%	2769, 20.0%	3941, 29.4%	
Residence (*n*, %)				<0.0001
Community dwelling	6469, 95.2%	12,158, 87.8%	8196, 61.2%	
Institution	328, 4.8%	1696, 12.2%	5193, 38.8%	
Urbanization level (*n*, %)				<0.0001
Rural	914, 13.5%	2106, 15.2%	2084, 15.6%	
Suburban	2320, 34.1%	5117, 36.9%	5522, 41.2%	
Urban	3563, 52.4%	6631, 47.9%	5783, 43.2%	
Clinical Dementia Rating				<0.0001
1	4255, 62.6%	5715, 41.3%	1969, 14.7%	
2	2209, 32.5%	6054, 43.7%	4727, 35.3%	
≥3	333, 4.9%	2085, 15.1%	6693, 50.0%	
WHODAS 2.0 (mean ± SD)				
Cognition (domain 1)	46.9 ± 24.7	56.6 ± 23.6	79.3 ± 22.3	<0.0001
Mobility (domain 2)	17.3 ± 18.6	45.1 ± 23.6	77.7 ± 22.9	<0.0001
Self-care (domain 3)	16.7 ± 18.3	32.4 ± 25.4	53.0 ± 35.1	<0.0001
Getting along (domain 4)	48.0 ± 28.6	60.8 ± 26.5	83.6 ± 22.1	<0.0001
Life activities (domain 5)	53.3 ± 35.4	70.2 ± 34.4	85.6 ± 32.0	<0.0001
Participation (domain 6)	28.3 ± 19.5	40.1 ± 21.8	56.7 ± 25.2	<0.0001
Summary	34.4 ± 17.0	49.7 ± 18.2	71.5 ± 17.9	<0.0001

CDR = Clinical Dementia Rating. CDR result—zero (no dementia), 0.5 (questionable dementia), one (mild dementia), 2 (moderate dementia), and 3 (severe dementia).

**Table 2 ijerph-16-02168-t002:** Odds ratios of nonambulatory status among participants with different demographic backgrounds.

	β	Odds Ratio	95% Wald Confidence Limits		*p* Value
Intercept	−2.1371				<0.0001
Sex (ref = Female)					
Male	0.0144	1.014	0.963	1.069	0.5910
Age (years) (ref = 65–74)					
75–84	0.3239	1.383	1.298	1.473	<0.0001
≥85	0.7842	2.191	2.016	2.380	<0.0001
Residence (ref = Community dwelling)					
Institution	1.3374	3.809	3.576	4.058	<0.0001
Urbanization level (ref = Urban)					
Rural	0.0742	1.077	0.999	1.161	0.0538
Suburban	0.1479	1.159	1.097	1.225	<0.0001
CDR (ref = 1)					
2	0.8972	2.453	2.305	2.610	<0.0001
≥3	2.2822	9.798	9.138	10.506	<0.0001

CDR = Clinical Dementia Rating.

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
