# Peer review of "Association between Ambulatory Status and Functional Disability in Elderly People with Dementia"

_ijerph, 2019, doi:10.3390/ijerph16122168_

Round 1

Reviewer 1 Report

This is a clearly presented and cogent paper. The abstract offers a very lucid summary of the research, The background and materials section provide a very sound platform for the results. The findings are also lucidly conveyed. Overall, the paper is well structured and the standard of writing is high.

The paper is based upon secondary research with data drawn from a databank, but there are places in the research tools section where the writing seems to imply primary research was undertaken. A minor rewrite would avoid this issue. For example, under 2.2.2 it is stated that "We administered the 36-item Chinese version of WHODAS..." If I am following the process correctly, this tool was utilised rather than administered. Likewise, a slight rewrite under 2.2.3 would render it clearer that walk status has been previously evaluated in the tool being drawn upon (rather than being undertaken as a primary data collection method in this study).

The key area where I felt the paper could be enhanced is the discussion and conclusion. This would not require extensive changes, but some additional material could be advantageous. It does seem as though the discussion could be more evaluative. It is presently a very descriptive account of the findings. For example, it appears to be a significant finding that a nonambulatory status was significantly associated with the risk of institution residency. This finding is only reiterated and there is no evaluation of its implications. This finding warrants some additional scrutiny. Does anything need to be done to explore it further?

The conclusion could be more positively presented and underscore the key findings and their implications. There is always more research that can be taken! How can the findings from this research inform and shape future studies? (The abstract seems to offer a more positive statement on the conclusion of the study, than the conclusion itself.)

Overall, the quality of writing in the paper is high. There are some places, however, where I feel the expression can be adjusted:

For example, he term 'senile dementia' is used under 2.2.1. This is now a contested term, as it implies an intrinsic association between ageing and dementia. Is it possible to remove the work 'senile'?

It is a pedantic point, but at the top of page 8 the term 'relatively unique' is used. 'Unique' is an absolute term, so can't be relative. I suggest using another term such as 'distinctive'. This sentence also leaves the reader wondering what is unique (or distinctive) about services in Taiwan. A small degree of elaboration here would satisfy the reader's curiosity. 

There is also a slight loss of logic in the sentence: "the study analyzed disability in patients with dementia in Taiwan, and thus, we did not analyze patients with mild cognitive impairment." (p.7). I would remove 'in Taiwan' from this sentence.

A small degree of elaboration could also be applied to the final sentence of the discussion. The intention of the authors is fairly clear here, but the reader might be left requiring a little more detail on what is being asserted with regard to living status (e.g. family home or residential care)?

Author Response

Point 1: The paper is based upon secondary research with data drawn from a databank, but there are places in the research tools section where the writing seems to imply primary research was undertaken. A minor rewrite would avoid this issue. For example, under 2.2.2 it is stated that "We administered the 36-item Chinese version of WHODAS..." If I am following the process correctly, this tool was utilised rather than administered. Likewise, a slight rewrite under 2.2.3 would render it clearer that walk status has been previously evaluated in the tool being drawn upon (rather than being undertaken as a primary data collection method in this study).

Response 1 Following your comment, we have revised it as following “The 36-item Chinese version of WHODAS 2.0 was utilised’’

Point 2: The key area where I felt the paper could be enhanced is the discussion and conclusion. This would not require extensive changes, but some additional material could be advantageous. It does seem as though the discussion could be more evaluative. It is presently a very descriptive account of the findings. For example, it appears to be a significant finding that a nonambulatory status was significantly associated with the risk of institution residency. This finding is only reiterated and there is no evaluation of its implications. This finding warrants some additional scrutiny. Does anything need to be done to explore it further?

Response 2 Following your comment, we have revised it as following

Discussion

Our study found that the severity of dementia was a valuable predicting marker associated with walking status. The appearance of abnormalities of gait increase the risk of dementia. Both of them were common with increasing age and making these two problems a major issue of public health. Patients with dementia with a higher CDR were associated with nonambulatory status in the getting along, life activities, and participation domains. Thus, prevention of dementia progression may help maintain walking status in patients with dementia. Furthermore, maintaining walking status and preventing nonambulatory status in patients with dementia can effectively improve cognitive function [25, 26]. How to maintain the function of gait and decrease the severity of dementia is an urgent problem and require more research.

Point 3: The conclusion could be more positively presented and underscore the key findings and their implications. There is always more research that can be taken! How can the findings from this research inform and shape future studies? (The abstract seems to offer a more positive statement on the conclusion of the study, than the conclusion itself.)

Response 3 Following your comment, we have revised it as following :

Conclusion

Our research provides clinical features for elderly people with gait disorders and dementia. The quality of the gait can be sufficiently reliable and responsive to measuring the severity of dementia. Poor gait ability was associated with a higher CDR and WHODAS 2.0 score in individuals with dementia. Nonambulatory status was associated with both greater severity of dementia and functional disability. Further research involving longitudinal studies is certainly warranted to investigate the causes in detail.

Point 4: For example, he term 'senile dementia' is used under 2.2.1. This is now a contested term, as it implies an intrinsic association between ageing and dementia. Is it possible to remove the work 'senile'?

Response 4 Following your comment, we remove the term 'senile '.We have revised it as following “The CDR, a global rating scale, was developed for a prospective study on mild dementia of the Alzheimer type. Its reliability, validity, and correlational data have been examined previously. Six cognitive categories are involved in this international standard for the staging of dementia, namely memory, orientation, judgment and problem solving, community affairs, home and hobbies, and personal care. The CDR was found to unambiguously distinguish between severely impaired and healthy older adults.’’

Point 5: It is a pedantic point, but at the top of page 8 the term 'relatively unique' is used. 'Unique' is an absolute term, so can't be relative. I suggest using another term such as 'distinctive'. This sentence also leaves the reader wondering what is unique (or distinctive) about services in Taiwan. A small degree of elaboration here would satisfy the reader's curiosity. 

Response 5 Following your comment, we have revised it as following “Taiwan's health insurance, medical, and social welfare systems are relatively distinctive.’’

Point 6: There is also a slight loss of logic in the sentence: "the study analyzed disability in patients with dementia in Taiwan, and thus, we did not analyze patients with mild cognitive impairment." (p.7). I would remove 'in Taiwan' from this sentence.

Response 6 Thank you for your valuable comments. Following your comment, we have revised it as following “Taiwanese people with dementia’’.

Point 7: A small degree of elaboration could also be applied to the final sentence of the discussion. The intention of the authors is fairly clear here, but the reader might be left requiring a little more detail on what is being asserted with regard to living status (e.g. family home or residential care)?

Response 7 Following your comment, we have added ‘’family’’ in the sentense. We have revised it as following “In addition, elderly people in Taiwan mostly live at family home; therefore, the results of this study may not be applicable to patients in other countries.’’

Reviewer 2 Report

Association Between Ambulatory Status and Functional Disability in Elderly People with Dementia

Thank for sending your paper to the International Journal of Environmental Research and Public Health, and providing me with the opportunity to review your work.

An interesting and important topic, I have some specific comments below, I hope you find these constructive, as this is my intention.

Abstract:

-          Your first sentence is grammatically incorrect which impacts on the understanding, please amend

-          I am unsure what you mean by ‘… which is essential to the progression of dementia.’ Please clarify this comment

-          The conclusion needs to be amended slightly as cause and effect has not been proven in this cross-sectional study

Introduction

-          The third paragraph is only one sentence long, this is not a paragraph – this information should be expanded upon or amalgamated into one of the other paragraphs

Materials and methods

-          Clear and precise provision of information

Results

-          Clear and precise provision of information

-          Graphs require a clear heading and an explanation under the first graph explaining CDR1, 2 and 3 would support clarity

Discussion

-          Limitations should be presented in a standalone paragraph

-          The implications of your results with wider research could be further expanded

Conclusion

-          This element needs to be expanded further with clear implications of the findings of your work

Author Response

Point 1: Abstract:

-          Your first sentence is grammatically incorrect which impacts on the understanding, please amend

Response 1 Following your comment, we have revised it as following “Dementia is highly comorbid with gait disturbance, and both conditions negatively impact the ability of elderly people to conduct daily living activities.”

-          I am unsure what you mean by ‘… which is essential to the progression of dementia.’ Please clarify this comment

Response 1 Following your comment, we have revised it as following “The ambulatory status of older adults with dementia may cause variable functional disability, which is crucial for the progression of dementia..”

-          The conclusion needs to be amended slightly as cause and effect has not been proven in this cross-sectional study

Response 1 Following your comment, we have revised it as following “In conclusion, the present study clearly demonstrated the role of ambulatory status in functional disability in older adults with dementia, and the association persisted among older adults of different ages and severities of dementia.’’

Point 2: Introduction

-          The third paragraph is only one sentence long, this is not a paragraph – this information should be expanded upon or amalgamated into one of the other paragraphs

Response 2 Following your comment, we have revised it and combined the sentence to the front paragragh.

Point 3: Results

-          Graphs require a clear heading and an explanation under the first graph explaining CDR1, 2 and 3 would support clarity

Response 3 Following your comment, we have revised it as following “CDR = Clinical Dementia Rating. CDR result - zero (no dementia), 0.5 (questionable dementia), one (mild dementia), 2 (moderate dementia), and 3 (severe dementia).’’

Point 4: Discussion

-          Limitations should be presented in a standalone paragraph

Response 4 Following your comment, we have revised it into a standalone paragraph.

-          The implications of your results with wider research could be further expanded

Response 4 Following your comment, we have revised it as following:

Our study found that the severity of dementia was a valuable predicting marker associated with walking status. The appearance of abnormalities of gait increase the risk of dementia. Both of them were common with increasing age and making these two problems a major issue of public health. Patients with dementia with a higher CDR were associated with nonambulatory status in the getting along, life activities, and participation domains. Thus, prevention of dementia progression may help maintain walking status in patients with dementia. Furthermore, maintaining walking status and preventing nonambulatory status in patients with dementia can effectively improve cognitive function [25, 26]. How to maintain the function of gait and decrease the severity of dementia is an urgent problem and require more research.

Point 5: Conclusion

-          This element needs to be expanded further with clear implications of the findings of your work

Response 5 Following your comment, we have revised it as following “Our research provides clinical features for elderly people with gait disorders and dementia. The quality of the gait can be sufficiently reliable and responsive to measuring the severity of dementia. Poor gait ability was associated with a higher CDR and WHODAS 2.0 score in individuals with dementia. Nonambulatory status was associated with both greater severity of dementia and functional disability. Further research involving longitudinal studies is certainly warranted to investigate the causes in detail.

’’
